# Cross-Sectional and Longitudinal Association between Glycemic Status and Body Composition in Men: A Population-Based Study

**DOI:** 10.3390/nu10121878

**Published:** 2018-12-03

**Authors:** Khaleal Almusaylim, Maggie Minett, Teresa L. Binkley, Tianna M. Beare, Bonny Specker

**Affiliations:** EA Martin Program in Human Nutrition, South Dakota State University, SWC, Box 506, Brookings, SD 57007, USA; khalealabdulha.almusaylim@jacks.sdstate.edu (K.A.); maggie.minett@sdstate.edu (M.M.); teresa.binkley@sdstate.edu (T.L.B.); tianna.beare@sdstate.edu (T.M.B.)

**Keywords:** prediabetes, type 2 diabetes, total body fat, total body lean, appendicular fat, appendicular lean, body composition, cohort study

## Abstract

This study sought to evaluate the associations between changes in glycemic status and changes in total body (TB), trunk, and appendicular fat (FM) and lean mass (LM) in men. A population-based study of men aged 20–66 years at baseline were included in cross-sectional (*n* = 430) and three-year longitudinal (*n* = 411) analyses. Prediabetes was defined as fasting glucose 100–125 mg/dL. Type 2 diabetes (T2D) was determined by: self-reported diabetes, current anti-diabetic drug use (insulin/oral hypoglycemic agents), fasting glucose (≥126 mg/dL), or non-fasting glucose (≥200 mg/dL). Body composition was evaluated by dual-energy X-ray absorptiometry. Longitudinal analyses showed that changes in TB FM and LM, and appendicular LM differed among glycemic groups. Normoglycemic men who converted to prediabetes lost more TB and appendicular LM than men who remained normoglycemic (all, *p* < 0.05). Normoglycemic or prediabetic men who developed T2D had a greater loss of TB and appendicular LM than men who remained normoglycemic (both, *p* < 0.05). T2D men had greater gains in TB FM and greater losses in TB and appendicular LM than men who remained normoglycemic (all, *p* < 0.05). Dysglycemia is associated with adverse changes in TB and appendicular LM.

## 1. Introduction

Prediabetes and type 2 diabetes (T2D) are major public health issues in the United States. The national prevalence of prediabetes and T2D among adults aged ≥20 years has increased over time, with the prevalence of prediabetes rising from 26% in 1988–1994 [1] to 37% in 2009–2012 [2]. Recent estimates from the Centers for Disease Control and Prevention indicate that 15–30% of prediabetic cases progress from impaired fasting glucose or impaired glucose tolerance to T2D within five years. There are currently 25.8 million adults in the United States with prediabetes that will develop T2D by 2020, which will double the number of individuals affected by T2D [3]. The prevalence of T2D rose from 7% in 2005 to 12% in 2011 [4], and is projected to increase 165% by 2050 [5].

Case-control and cross-sectional studies have reported inconsistent associations of total body (TB), trunk, and appendicular fat mass (FM) and lean mass (LM) with prediabetes and T2D diabetes in middle-aged and older adults, including a positive association [6], an inverse association [7,8,9,10], and no association [6,8,9]. Prospective studies aimed at investigating the relationship between baseline glycemic status and subsequent changes in TB and regional distribution of FM and LM are sparse and inconclusive. Some studies reported differences in body composition measurements among glycemic groups [11,12,13,14], but others did not find differences [15,16]. The possible explanation for inconsistent findings includes different durations of follow-up period, sample size, and other covariates that were not adjusted for when investigating the association between various measures of body composition and glycemic status. We found no epidemiological studies that investigated the association between glycemic status (men with prediabetes who revert to normoglycemia, or men who are normoglycemic or prediabetic at baseline and convert to T2D) and changes in the TB and regional distribution of FM and LM.

The objective of the present analysis was to examine the associations between baseline and changes in glycemic status with baseline and changes over three years in TB, trunk, and appendicular FM and LM. The following *a priori* hypotheses were tested: (1) men with prediabetes or T2D at baseline would have higher TB and trunk fat measurements but lower appendicular fat than normoglycemic men; (2) men who were normoglycemic at baseline and developed either prediabetes or T2D would have increases in TB and trunk FM, and decreases in TB and appendicular LM over the three-year study compared with men who remained normoglycemic; (3) among men with prediabetes at baseline, the changes in TB and trunk FM and LM would differ over the three-year study depending on whether they remained prediabetic or developed T2D versus reverting to a normoglycemic state. Those who develop T2D would gain TB FM and lose LM, while those reverting to normoglycemia would lose FM and maintain or gain LM; and (4) men with T2D at baseline would have decreases in TB and trunk LM over the three-year study compared to normoglycemic men.

## 2. Materials and Methods

### 2.1. Study Population

The South Dakota Rural Bone Health Study is a population-based longitudinal study designed to investigate the impact of lifestyle factors on bone and body composition. The design and rationale of the study have been described elsewhere [17]. Briefly, adults aged 20 to 66 years, from eight counties in eastern South Dakota, were eligible for enrollment. A total of 1271 participants were recruited between 2001–2004 (baseline), and followed for an average of 3.0 years (range of 2.8 to 3.8 years), and the current analysis was limited to men (*n* = 544). Among those participants, 410 men farmed at least 75% of their lives (rural) and 134 men never lived on an active farm (non-rural). The rural population was divided into Hutterites and non-Hutterites. A Hutterite was defined as a participant of Hutterite descent who resided on a Hutterite colony. Hutterites are an Anabaptist religious group who believe in isolated communal living and self-sufficiency through an agriculturally advanced lifestyle. Non-Hutterites were randomly selected from the eight-county region as described elsewhere [17].

Men with chronic use (> six months) of immunosuppressants, anticonvulsants, or steroids or a diagnosis of type 1 diabetes mellitus at baseline were not eligible for inclusion in the original cohort. For baseline analyses, we excluded men with missing glucose measures at either baseline (*n* = 12) or follow-up (*n* = 34), baseline body composition measurements (*n* = 23), or men who withdrew from the study (*n* = 45) (Figure 1). For follow-up analyses, we further excluded men who did not have body composition measurements at follow-up (*n* = 19). These exclusions led to 430 men in the baseline analyses and 411 in the follow-up analyses. The study was approved by the South Dakota State University Institutional Review Board (IRB#1406004), and informed consent was obtained from all of the participants.

### 2.2. Assessment of Covariates

Questionnaires were administered at study enrollment and at three years to obtain information on demographic and lifestyle characteristics as well as quarterly physical activity and dietary intake recalls. Information on smoking status and specific details regarding the use of prescription drugs was not collected at baseline; however, an 18-month survey was used to obtain this information. Participants were asked to provide information on types of smoking, such as cigarettes, cigars, and pipes, and were classified as smokers or non-smokers. The presence or absence of hypertension at 18 months was based on self-reported information on the use of antihypertensive medication. 

#### 2.2.1. Anthropometric Measures

Body height and weight were measured in lightweight clothes without shoes using a calibrated stadiometer and scale. Standing height was measured to the nearest 0.5 cm in duplicate with a stadiometer (Seca, Chino, CA, USA). A third measurement was taken if the discrepancy between the duplicate height measurements was more than 0.5 cm. Weight was measured to the nearest 0.1 kg with a digital scale (Seca Model 770, Chino, CA, USA).

#### 2.2.2. Physical Activity Assessment

The Paffenbarger Physical Activity Questionnaire (PPAQ) was used to measure the average amount of time spent in sedentary behaviors and different intensity levels of physical activity during the past week [18]. Participants were asked to recall how many hours on their usual weekday and weekend day they spent sleeping, sitting, and engaging in moderate or vigorous intensity activity. Since the time spent in sleeping, sitting, and participating in moderate plus vigorous activity was measured, the remaining time was considered light activity. The PPAQ was administered quarterly over the first three years of the study. To properly report participants’ physical activity, trained personnel administered the PPAQ by interview. The average time spent in sitting and moderate-plus-vigorous activity, as well as the average sleeping time, was calculated. The validity and reliability of the PPAQ have been established to measure physical activity intensities in men [19] and rural populations [20].

#### 2.2.3. Dietary Assessment

Dietary intake was assessed using 24-h dietary recalls that were conducted at similar times as the activity recall. Trained interviewers administered 24-h dietary recalls, and dietary recall data were analyzed using Nutritionist Pro software (version 2.3.1, 2004, First DataBank, Inc., San Bruno, CA, USA) to estimate macronutrient and micronutrient intakes. For foods not available in the Nutritionist Pro software, the nutrient composition of the foods was obtained from recipes and entered into the diet analysis software. Activity levels and nutrient intakes at baseline were the averages of the baseline, three-month and six-month recalls, and measures at the 36-month visit were the averages of the 30-month, 33-month, and 36-month recalls.

### 2.3. Ascertainment of Glycemic Status

According to American Diabetes Association classifications, individuals with a fasting blood glucose of 100 mg/dL to 125 mg/dL were classified as having prediabetes [21]. T2D was determined by one of the following criteria: self-reported T2D, current use of an anti-diabetic drug (insulin or oral hypoglycemic agents), or a fasting blood glucose concentration ≥126 mg/dL or a non-fasting blood glucose concentration ≥200 mg/dL [21]. The same criteria for the diagnosis of prediabetes and T2D were applied at both the baseline and three-year visits. Attempts were made to obtain fasting blood samples at each visit, and measurements were made in the field from a sample of venous whole blood (with added ethylenediaminetetraacetic acid) using an Accu-Check Advantage glucometer (Roche Diagnostics, Indianapolis, IN, USA). 

### 2.4. Body Composition Measurements

Body composition was assessed using dual-energy x-ray absorptiometry (DXA) (Discovery, Software Version 12.01, Hologic, Waltham, MA, USA). A TB scan was completed with boundaries for the various anatomical regions set according to manufacturer’s specifications. The step phantom scan for body composition calibration was completed weekly as suggested by the manufacturer. Prior to DXA measurements, the Hologic spine phantom was scanned for quality control. All of the scans were analyzed by the same technician who was certified by the International Society of Clinical Densitometry. Scan results were deleted for obese participants with an equivalent epoxy thickness greater than 12 inches, as determined by the Hologic software, per manufacturer recommendations (*n* = 16 men: *n* = seven Hutterite, *n* = four rural, *n* = five non-rural). DXA-derived measurements of TB, trunk and appendicular FM and LM were expressed in kilograms. Our coefficients of variation for TB FM and LM assessed in 15 adults (one male) using triplicate scans with repositioning between each scan are <1.5%.

### 2.5. Statistical Analysis

All of the continuous variables were tested for normality before performing the analyses. Analysis of variance adjusting for multiple comparisons for continuous variables and a Fisher’s exact test for categorical variables were used to determine statistical significance in baseline characteristics among the glycemic categories. The annual absolute change in each body composition measure was calculated as the follow-up value minus baseline value divided by length of follow-up in years. 

Multiple regression models were used to estimate marginal means ± standard error of the mean (sem) for baseline body composition parameters and changes in outcome measures by different categories of glycemic status. Differences in marginal means among glycemic groups were evaluated using post hoc contrast tests based on the hypotheses. A priori determined covariates (age at baseline, height, population group, percent time in moderate-plus-vigorous activity, and total daily caloric intake) were included in all of the models, since they were found to be associated with at least one body composition measure. The multiple regression models that were used for baseline analyses included these covariates, and the FM model included the LM of the same compartment (TB, trunk, or appendicular), and the LM model included covariates plus the FM of the same compartment. Multiple regression models for the longitudinal analyses were adjusted for the same covariates, as well as changes in percent time in moderate-plus-vigorous activity and total daily caloric intake between baseline and three-year follow-up, baseline measure of the specific body compartment (TB, trunk, or appendicular), and baseline and annual changes in the FM or LM of same compartment. Due to issues with multicollinearity and the problem of body composition measures being components of both body mass index (BMI) and weight, neither BMI nor weight was included as covariates. The assumptions of linearity, normality, and homoscedasticity were evaluated visually to ensure no violation of assumptions. All of the analyses were performed using JMP software (version 13, SAS Institute, Cary, NC, USA), and the statistical significance level was set at *p* < 0.05 (two-tailed). 

## 3. Results 

### 3.1. Subject Characteristics

At the baseline visit, 358 (83.2%) of the men were normoglycemic, 51 (11.9%) were prediabetic based on fasting glucose concentrations, and 21 (4.9%) had T2D (14 self-reported a medical diagnosis, six based on fasting glucose, and one based on non-fasting glucose). Of the 345 men who were normoglycemic at baseline and for whom glucose and body composition measurements were available at three years, 272 (78.9%) remained normoglycemic, 65 (18.8%) progressed to prediabetes, and eight (2.3%) progressed to T2D (five based on fasting glucose, one based on non-fasting glucose, one self-reported a medical diagnosis, and one taking anti-diabetic medication). Among the 48 prediabetic men, 19 (39.6%) remained prediabetic, 25 (52.1%) reverted to normoglycemic, and four (8.3%) progressed to T2D based on fasting glucose concentrations. Of the 18 T2D men who were diabetic throughout the study, four (22.2%) self-reported a medical diagnosis, 11 (61.1%) were taking anti-diabetic medication, and three (16.7%) had T2D based on fasting glucose concentrations. 

Participant characteristics by glycemic categories at baseline and follow-up are summarized in Table 1 and Table 2. The mean age (+ sem) was 42.7 ± 0.6 years (range: 20 to 66 years), and men with T2D at baseline were older than those who were normoglycemic. The study population was 37.4% Hutterite, 37.2% rural non-Hutterite, and 25.4% non-rural. Hutterites and married men had a higher prevalence of prediabetes and T2D than non-Hutterites and single men. Men with prediabetes weighed more than normoglycemic men. A higher percentage of prediabetic and T2D men were taking antihypertensive medication than normoglycemic men, and carbohydrate intake was greater in normoglycemic men than in prediabetic and T2D men. At follow-up, men with T2D had lower caloric and carbohydrate intake than men who remained normoglycemic throughout the study or men who were normoglycemic and developed prediabetes (Table 2). Prediabetic men who remained prediabetic at three years increased their carbohydrate intake, and men who remained normoglycemic throughout the study increased their weight, time spent in moderate-plus-vigorous activity, and average daily fat intake. The overall mean changes over the three-year study in percent time in moderate-plus-vigorous activity and average dietary intake of calories, carbohydrates, fat, and protein were not significant (mean changes were 0.6 ± 0.4%, 10 ± 28 kcal, 1.0 ± 3.8 g, 3.0 ± 1.6 g, and 0.0 ± 1.5 g, respectively), and changes in caloric and macronutrient intakes did not differ by glycemic categories (data not shown). 

### 3.2. Cross-Sectional Assessment of Baseline Body Composition in Men with Prediabetes or Type 2 Diabetes 

There were no differences in TB, trunk, or appendicular FM or LM among the three glycemic groups at baseline when covariates were included in the analyses (Table 3). Prediabetic men weighed more than normoglycemic men. 

### 3.3. Association between Development of Prediabetes and Type 2 Diabetes and Changes in Body Composition

Changes in body composition for the six glycemic groups are shown in Figure 2 and Figure 3. There were differences among men who developed prediabetes or T2D and men who remained normoglycemic regarding the annual change in TB and appendicular LM. Among men who were normoglycemic at baseline, those who progressed to prediabetic lost more TB and appendicular LM than those who remained normoglycemic (Figure 3). Normoglycemic or prediabetic men who developed T2D also had greater losses in TB and appendicular LM than men who remained normoglycemic (Figure 3). 

### 3.4. Changes in Body Composition among Prediabetic Men Depending on Reversion to Normoglycemia

There were no differences in the FM or LM between prediabetic men who remained prediabetic and those that reverted to normoglycemia (Figure 2 and Figure 3). In general, prediabetic men who reverted to normoglycemia had negative changes in FM, whereas men who remained prediabetic had positive changes.

### 3.5. Changes in Body Composition among Type 2 Diabetic Men 

Men who were T2D at baseline had greater gains in TB FM (Figure 2) and greater losses in TB and appendicular LM (Figure 3) than men who remained normoglycemic over the three-year follow-up. 

## 4. Discussion

This is the first prospective population-based cohort study investigating the association of baseline glycemic status and changes in glycemic status over time with changes in TB, trunk, and appendicular FM and LM. Consequently, the findings of prior observational longitudinal studies cannot be compared to our findings. The findings of the current study indicate that there were no baseline differences among glycemic groups in TB, trunk, or appendicular FM and LM. Normoglycemic men who developed prediabetes or T2D had greater losses in TB and appendicular LM than men who remained normoglycemic. Men who had T2D throughout the study period had greater gains in TB FM, and greater losses in TB and appendicular LM, than men who were normoglycemic throughout the study. No differences were observed in changes in weight or body composition measures among prediabetic men who reverted to normoglycemia compared to those who remained prediabetic.

Contrary to our first hypothesis, we did not find differences in TB and regional FM and LM at baseline among the glycemic groups. The present study differs from other studies [6,7,8,10,22,23] due to the adjustment for covariates and inclusion of other body composition compartments (e.g., when determining whether TB FM was associated with glycemic status, TB LM was included in the statistical model). The inclusion of these covariates resulted in the relationships becoming non-significant. 

Consistent with our hypothesis, normoglycemic men who developed prediabetes or T2D had greater losses in TB and appendicular LM than men who remained normoglycemic, but we found no association with changes in FM. A positive association between glucose concentrations and intermuscular adipose tissue has been reported [24], and it has been suggested that hyperglycemia stimulates the differentiation of mesenchymal stem cells derived from adipose and muscle tissues into adipocytes by activating the protein kinase C β pathway [25]. Other studies also have reported an association between hyperglycemia and reduced TB and appendicular LM in men [10]. The underlying mechanisms of the decline in LM may include elevated circulating concentrations of inflammatory markers and oxidative stress. Biomarkers of inflammation, tumor necrosis factor alpha, and C-reactive protein stimulate the loss of skeletal muscle through the activation of nuclear factor kappa B [26] and the inhibition of protein synthesis [27]. Oxidative stress contributes to a catabolic and anabolic imbalance in skeletal muscle, mitochondrial damage, and muscle atrophy and apoptosis [28]. These findings indicate that elevated inflammation markers in the presence of oxidative stress in prediabetic and T2D men can induce the loss of TB and appendicular LM. The present findings from the study support these reports. 

The association of changes in prediabetes status over time with changes in body composition by compartments have not been previously investigated. A few prospective studies have examined changes in TB and appendicular FM and LM in prediabetic individuals compared to normoglycemic controls. Our findings are similar to other studies reporting no difference in longitudinal changes in TB and lower extremity LM between individuals with and without prediabetes [14,16,29]. On the contrary, others have reported a loss in TB FM and appendicular LM that was greater in prediabetics compared to their normoglycemic counterparts [14,29]. 

In addition to greater gains in TB FM among men with T2D than among normoglycemic men, we found a significant loss in TB and appendicular LM among men who either had T2D at baseline or developed T2D during the study compared to men who remained normoglycemic. By contrast, Park et al. [15] reported no differences in longitudinal changes in TB and appendicular FM and LM between older adults with normoglycemia and those diagnosed T2D. Our findings are consistent with longitudinal studies that have found T2D men gained TB FM and lost TB and appendicular LM [12,13,14]. The mechanism for fat gain and muscle loss may stem from insulin resistance in T2D. An excessive influx of free fatty acids into the systemic circulation resulting from the adipose tissue contributes to insulin resistance by increasing fat accumulation in the liver leading to decreased insulin clearance, and increasing fat accumulation in skeletal muscle by impairing glucose transport, decreasing muscle protein synthesis, and inducing muscle protein breakdown, leading to a reduced muscle surface area and insulin signaling [30,31].

The strengths of our study include the first prospective population-based study of the association between changes in glycemic status and changes in TB and regional body composition measured by DXA, our low dropout rate, and our statistical adjustment for the same body composition compartments. Our study has several limitations. The present study included predominantly white men, and our findings may not be generalizable to women or other races. The majority of the men were farmers who may have different activity patterns and dietary intake than non-farmers, which may influence the relationship between dysglycemia and body composition. However, a study conducted on a representative sample of the United States (U.S.) population reported a similar association between dysglycemia and reduced lean mass [10]. Another limitation is the sample size of some of the glycemic categories. We did not observe differences between those men who were prediabetic at baseline and remained prediabetic, or reverted to normoglycemia as we hypothesized. It is likely that our sample size (*n* = 19 and 25, respectively) was too small. Based on the observed means and standard deviations in changes in TB FM, we estimate that 72 men per group would be needed (*α* = 0.05, *β* = 0.20). Despite the small sample size in some categories, we did observe other differences that we hypothesized, including differences in changes in TB and appendicular LM between normoglycemic men who developed prediabetes or T2D and those who remained normoglycemic throughout the study. We relied on participants’ recall of diagnosis of T2D, antidiabetic medication usage, dietary intake, and physical activity. A self-reported diagnosis of T2D or the use of antidiabetic medication can lead to misclassification due to recall or reporting errors. Dietary and physical activity recalls may result in overestimation or underestimation. However, dietary intake and physical activity assessments were performed quarterly to consider seasonality. The 24-h diet recall [32] and PPAQ [19] are valid measures of dietary intake and physical activity in adults. Although this was a longitudinal study, given the period of time between measurements (three years), it is not possible to determine whether changes in glycemic status preceded changes in body composition or vice versa. Future studies should be over longer periods of time with more frequent measures of glycemic status and body composition in order to determine which factor comes first: dysglycemia or body composition changes. Only one fasting or non-fasting blood glucose measurement per visit was obtained for defining prediabetes and T2D. Although the American Diabetes Association recommends different criteria for screening for prediabetes and T2D using glycated hemoglobin, fasting blood glucose, and two-hour plasma glucose after an oral glucose tolerance test [21], numerous studies have reported that using two-hour plasma glucose test detects more cases of prediabetes and diabetes than using glycated hemoglobin or a fasting blood glucose test [33,34,35,36,37,38]; thus, we might have missed men with prediabetes and T2D using only one fasting blood glucose measurement, which would have made it more difficult to identify group differences in changes in body composition.

## 5. Conclusions

In conclusion, (1) there were no differences among glycemic groups in baseline TB and regional distribution of FM and LM; (2) men who were normoglycemic at baseline and developed prediabetes or T2D had greater losses in TB and appendicular LM than men who remained normoglycemic; (3) there were no differences in changes in body weight or composition among men who were prediabetic at baseline and remained prediabetic compared to those who reverted back to normoglycemia; and (4) men who had T2D at baseline had greater gains in TB FM and greater losses in TB and appendicular LM than normoglycemic men. These findings add to a growing body of literature on the associations between changes in glycemic status and body composition.

## Figures and Tables

**Figure 1 nutrients-10-01878-f001:**
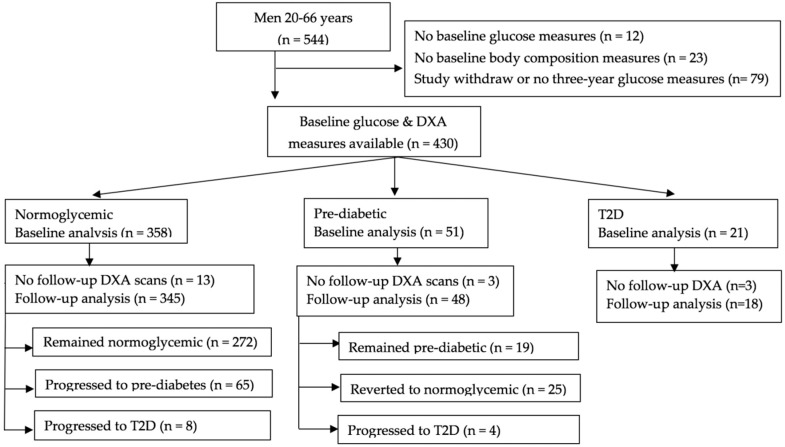
Flowchart of participants. Abbreviations: DXA, dual-energy X-ray absorptiometry; T2D, type 2 diabetic. Individuals who were unable to be categorized into glycemic groups were excluded.

**Figure 2 nutrients-10-01878-f002:**
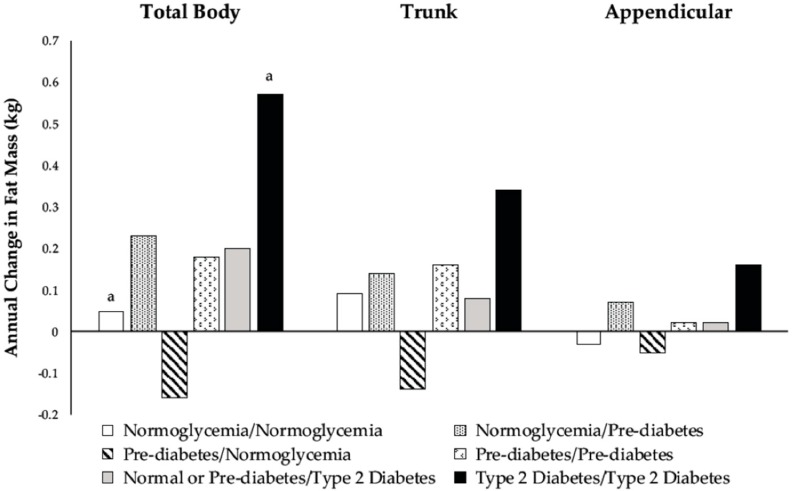
Adjusted marginal means of annual change from baseline in total body (*p* = 0.02), trunk (*p* = 0.06), and appendicular (*p* = 0.06) fat mass according to categories of glycemic status during the three-year follow-up. Model included baseline age, height, population group, percent time in moderate-plus-vigorous activity, average caloric intake, baseline measures of fat and lean mass in the specific body compartment (total body, trunk, or appendicular), changes in percent time in moderate-plus-vigorous activity and average caloric intake, and annual change in lean mass of the same compartment. Means with similar superscripts are different using post hoc contrast tests based on hypotheses.

**Figure 3 nutrients-10-01878-f003:**
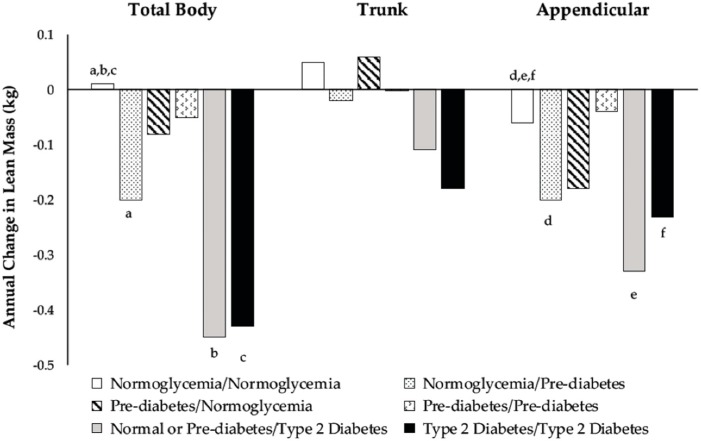
Adjusted marginal means of annual change from baseline in total body (*p* = 0.004), trunk (*p* = 0.24), and appendicular (*p* < 0.001) lean mass according to categories of glycemic status during the three-year follow-up. Model included baseline age, height, population group, percent time in moderate-plus-vigorous activity, average caloric intake, baseline measures of fat and lean mass in the specific body compartment (total body, trunk, or appendicular), changes in percent time in moderate-plus-vigorous activity and average caloric intake, and annual change in the fat mass of the same compartment. Means with similar superscripts are different using post hoc contrast tests based on hypotheses.

**Table 1 nutrients-10-01878-t001:** Baseline characteristics of the 430 men from the South Dakota Rural Bone Health Study cohort. NH: non-Hutterite.

	Normoglycemic	Prediabetic	T2D	*p*-Value ^1^
Participants (*n*)	358	51	21	
Demographics				
Age (years)	41.6 ± 0.6 ^a^	45.8 ± 1.7	53.0 ± 2.6 ^a^	<0.001
Population Group (%)				0.001
Hutterite (*n* = 161)	73.3	18.0	8.7	
NH Rural (*n* = 160)	89.3	8.8	1.9	
NH Non-rural (*n* = 109)	89.0	7.3	3.7	
Ever Married (%)	81.3	92.2	95.2	0.06
Anthropometrics				
Height (cm)	177.9 ± 0.4	177.9 ± 0.9	174.3 ± 1.5	0.06
Weight (kg)	91.1 ± 0.8 ^a^	98.5 ± 2.1 ^a^	95.9 ± 3.3	0.003
Lifestyle Variables				
Smokers (%)	33.2	24.0	38.1	0.43
BP Meds (%)	8.7	23.5	47.6	<0.001
% Time in MVPA ^2^	21.8 ± 0.5	23.1 ± 1.4	19.3 ± 2.1	0.32
Daily Macronutrient Intake ^2^				
Total energy (kcal)	2373 ± 33	2218 ± 87	2060 ± 135	0.03 ^3^
Carbohydrate (g)	265 ± 5 ^ab^	224 ± 12 ^b^	202 ± 19 ^a^	<0.001
Fat (g)	97 ± 2	98 ± 4	90 ± 7	0.55
Protein (g)	105 ± 2	102 ± 4	102 ± 7	0.80

Values are means ± sem or *n* (percentages). ^1^ Significance based on ANOVA for continuous variables and Fisher’s exact test for categorical variables; means with similar superscripts are different using a *post hoc* Tukey test. ^2^ Physical activity levels and nutrient intakes at baseline were the average of the baseline, 3-and 6-month recalls. ^3^ No means differed by post-hoc Tukey test for multiple comparisons. Abbreviations: T2D, type 2 diabetic; NH, non-Hutterite; BP Meds, hypertensive medications; MVPA, moderate-plus-vigorous physical activity.

**Table 2 nutrients-10-01878-t002:** Anthropometrics, activity levels, and diet intake of the 411 men from the South Dakota Rural Bone Health Study cohort, according to glycemic categories after three years of follow-up.

Baseline:	Normoglycemic		Prediabetic		Normoglycemic or Prediabetic	T2D	
Follow-Up:	Normoglycemic	Prediabetic	Normoglycemic	Prediabetic	T2D	T2D	*p*-Value ^1^
Participants (*n*)	272	65	25	19	12	18	
Baseline Age (years)	41.0 ± 0.7 ^ab^	42.6 ± 1.5 ^c^	43.9 ± 2.6	47.1 ± 2.3	51.9 ± 2.3 ^b^	53.0 ± 2.2 ^ac^	<0.001
Baseline Height (cm)	177.9 ± 0.4	177.0 ± 0.8	179.4 ± 1.0	176.9 ± 2.0	175.3 ± 1.7	175.1 ± 1.7	0.17
Weight (kg)							
Baseline	90.3 ± 0.9 ^‡^	92.5 ± 2.0	96.1 ± 3.3	98.8 ± 3.9	100.4 ± 4.1	96.0 ± 3.1	0.01 ^3^
Follow-Up	91.5 ± 0.9	93.3 ± 2.0	95.2 ± 3.4	100.1 ± 4.3	98.3 ± 4.0	96.0 ± 3.3	0.08
% Time MVPA ^2^							
Baseline	21.0 ± 0.6 ^‡^	23.7 ± 1.2	22.6 ± 1.7	24.4 ± 2.3	26.2 ± 3.0	19.8 ± 2.1	0.11
Follow-Up	22.5 ± 0.6	22.1 ± 1.1	19.9 ± 1.6	21.9 ± 2.1	24.5 ± 2.5	22.8 ± 2.4	0.77
Daily Intake ^2^							
Total Energy (kcal)							
Baseline	2344 ± 38	2435 ± 75	2284 ± 130	2140 ± 106	2248 ± 183	2067 ± 108	0.18
Follow-Up	2382 ± 38 ^a^	2386 ± 81 ^b^	2268 ± 102	2211 ± 95	2176 ± 189	1898 ± 132 ^ab^	0.02
Carbohydrate (g)							
Baseline	263 ± 5 ^a^	268 ± 11	239 ± 17	208 ± 14 ^‡^	233 ± 21	204 ± 15 ^a^	0.003
Follow-Up	265 ± 5 ^a^	267 ± 13 ^b^	248 ± 16	232 ± 14	208 ± 19	186 ± 16 ^ab^	0.001
Fat (g)							
Baseline	95 ± 2	102 ± 4	101 ± 7	94 ± 6	94 ± 10	91 ± 8	0.57
Follow-Up	101 ± 2 ^‡^	100 ± 3	95 ± 5	98 ± 6	99 ± 11	90 ± 8	0.75
Protein (g)							
Baseline	103 ± 2	107 ± 4	105 ± 7	96 ± 5	101 ± 11	101 ± 6	0.79
Follow-Up	104 ± 2	106 ± 3	105 ± 5	98 ± 5	107 ± 11	93 ± 8	0.50

Values are unadjusted means ± sem or *n* (percentages). ^‡^ Significant change from baseline to follow-up based on paired *t*-test. ^1^ Significance among glycemic categories based on ANOVA for continuous variables and Fisher’s exact test for categorical variables; means with similar superscripts are different using a post hoc Tukey test. ^2^ Physical activity (PA) levels and nutrient intakes at baseline were the averages of the baseline, three-month, and six-month recalls, and at follow-up were the averages of 30-month, 33-month, and 36-month recalls. ^3^ No means differed by post hoc Tukey test for multiple comparisons. Abbreviations: T2D, type 2 diabetic; MVPA, moderate-plus-vigorous physical activity.

**Table 3 nutrients-10-01878-t003:** Total body and regional body composition in the 430 men from the South Dakota Rural Bone Health Study cohort, according to glycemic status at baseline.

	Normoglycemic	Prediabetic	T2D	*p*-Value ^1^
Participants (*n*)	358	51	21	
Body Weight (kg)				
Unadjusted Model	91.1 ± 0.8 ^a^	98.5 ± 2.1 ^a^	95.9 ± 3.3	0.003
Basic Model ^2^	91.0 ± 0.8 ^a^	97.0 ± 3.2 ^a^	95.9 ± 3.2	0.01
Fat Mass (kg)				
Total Body				
Unadjusted Model	22.1 ± 0.5 ^ab^	26.4 ± 1.2 ^b^	26.7 ± 1.9 ^a^	0.001
Full Model ^3^	22.6 ± 0.3	24.0 ± 0.9	23.3 ± 1.5	0.38
Trunk				
Unadjusted Model	11.4 ± 0.3 ^ab^	14.2 ± 0.8 ^b^	15.1 ± 1.2 ^a^	<0.001
Full Model ^3^	11.8 ± 0.2	12.6 ± 0.5	11.8 ± 0.9	0.38
Appendicular				
Unadjusted Model	9.6 ± 0.2 ^a^	11.1 ± 0.5 ^a^	10.4 ± 0.8	0.01
Full Model ^3^	9.7 ± 0.2	10.3 ± 0.4	10.1 ± 0.7	0.66
Lean Mass (kg)				
Total Body				
Unadjusted Model	67.0 ± 0.4	69.9 ± 1.2	66.8 ± 1.8	0.07
Full Model ^4^	67.1 ± 0.3	67.9 ± 0.8	67.9 ± 1.2	0.56
Trunk				
Unadjusted Model	32.7 ± 0.2 ^a^	34.5 ± 0.6 ^a^	34.1 ± 0.9	0.01
Full Model ^4^	32.7 ± 0.1	33.1 ± 0.4	33.8 ± 0.6	0.24
Appendicular				
Unadjusted Model	30.5 ± 0.2	31.6 ± 0.6 ^a^	28.9 ± 0.9 ^a^	0.04
Full Model ^4^	30.5 ± 0.2	31.0 ± 0.4	30.3 ± 0.7	0.44

Data are means and marginal means ± sem. ^1^
*p*-values are from multiple regression models. Means with similar superscripts are different using a post hoc contrast test. ^2^ Basic model adjusted for age, height, population group, percent time in moderate-plus-vigorous activity, and average daily calories. ^3^ Fat mass models included covariates in basic model plus lean mass of same compartment (total body, trunk, or appendicular). ^4^ Lean mass models included covariates in basic model plus fat mass of same compartment. Abbreviations: T2D, type 2 diabetic.

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
