# Peer review of "Cross-Sectional and Longitudinal Association between Glycemic Status and Body Composition in Men: A Population-Based Study"

_nutrients, 2018, doi:10.3390/nu10121878_

Round 1
Reviewer 1 Report
Summary
The authors studied and compared the effect of changes in glycemic status on changes in total body, trunk and appendicular fat and lean mass. They showed that total body and appendicular fat mass and lean mass differed among glycemic groups.
Broad comments
The authors did a nice study with enough participants and collected enough data to present convincing and statistical reliable data. They show a correlation between body composition and glycemic status. But that overweight people have a higher risk for pre-diabetes and T2D is known and that a reduction of the weight reduces the risk also. The problem is, as they write themselves, they have only the data from the start and the end after 3 years. It would be interesting to see the time course of the development, what comes first, reduction of fat or change in glycemic status, is there somewhere a change in lifestyle, activity, calories intake, because the question why only a part of obeses people develops pre-diabetes or T2D is not fully clarified. As the paper is now, the scientific findings are little.
Specific comments
Line 11 – 14: Improve first two phrases of abstract.
Line 202, Table 2.: Not clear whether baseline characteristics are from the start, split accordingly to the glycemic results after 3 years, or whether they are the end data after 3 years.
Author Response
Point 1: Broad comment regarding inclusion of lifestyle factors:
We agree that inclusion of lifestyle factors may be important in understanding the development of T2D. It also is possible that lifestyle factors may result in differences, or explain differences, in total and regional body composition among glycemic groups. We included baseline percent time in moderate plus vigorous physical activity (% time MVPA) and average daily intake of total calories in the baseline models and these variables, along with changes in these variables over the three-year follow-up, in the longitudinal models. Smoking was not associated with any of the outcome measures and was therefore not included. The inclusion of these covariates in the fat and lean mass baseline models did not alter the conclusions, but annual changes in appendicular fat mass by dysglycemia group were no longer statistically different among glycemic categories (p=0.06 rather than p=0.05). Differences in annual changes in lean mass by glycemic categories were unchanged. We have incorporated these changes into the revised manuscript.
Point 2: Line 11-14: Improve first two phrases of abstract
We deleted the first two phrases from the abstract, which now states “This study sought to evaluate the associations between changes in glycemic status and changes in total body (TB), trunk, and appendicular fat (FM) and lean mass (LM) in men.” We also changed the last sentence in the abstract to “Dysglycemia is associated with adverse changes in TB and appendicular LM.”
Point 3: Line 202, Table 2:
We completely revised Table 2. Originally this table included only data at baseline, but now includes mean weight, activity levels and dietary intake at both baseline and follow-up. We felt this was would be more informative to the readers.
Reviewer 2 Report
This manuscript reports on the association between glycemic status and changes in body composition in adult men from the South Dakota Rural Bone Health Study. Data from 430 men were analyzed for associations between glycemic status and body composition at baseline, and data from 411 men were analyzed after 3 years of follow-up. Men for both analyses were categorized into one of three groups, normoglycemic, presence of pre-diabetes or presence of type 2 diabetes (T2D). Body composition was assessed by dual-energy x-ray absorptiometry, and total body (TB), trunk and appendicular fat mass (FM) and lean mass (LM) were assessed. Multivariate linear regression models were used to assess relationships for baseline body composition parameters and changes in outcome measures by the different categories of glycemic status. Results indicate that body composition was not associated with glycemic status at baseline. However, normoglycemic men who developed pre-diabetes gained more appendicular FM and lost more appendicular and TB LM than those who remained normoglycemic. Also, men who developed T2D during the 3-year follow-up lost more TB and appendicular LM compared to those who remained normoglycemic. Men with T2D gained more TB and appendicular FM and lost more TB and appendicular LM versus men who remained normoglycemic.
This manuscript addresses two topics of interest, glycemia and body composition, although assessing their associations is not novel. Regardless, the authors note that no epidemiological studies investigating the association between changes in glycemic status and changes in total body and appendicular body composition have been published. The authors conclude that dysglycemia adversely affects body composition, though I believe they intended to state that dysglycemia is adversely associated with body composition. Throughout the paper, the authors need to modify the language to make it clear that they are assessing associations and that causality cannot be established based on these results.
This manuscript is adequately written overall, but several significant items need to be addressed and are detailed in the Specific Comments to the Authors section.
Specific Comments to the Authors
General—throughout the manuscript, the authors refer to the results in a manner that indicates a causal relationship. For example, heading 3.3 is labelled, “The Effect of the Development of Pre-Diabetes and Type 2 Diabetes on Changes in Body Composition.” Reference to an effect also exists in the Abstract and Discussion sections. Since this is an epidemiological study, please refer to the observed results as associations.
Methods—The statistical analysis section states that age, height and population group were the only covariates accounted for in the basic model. Items such as education level, smoking status, physical activity, etc. may all be confounders. It is recommended that these, and other relevant variables, be added to the models to determine if an independent association still remains.
Methods/Discussion—The Discussion is the first mention of why the statistical analyses and models did not include body mass index (BMI). However, it is recommended that its exclusion be acknowledged in the Methods section with a brief reason for the exclusion.
Discussion—the study population is unique and likely not representative of the general U.S. population. A brief discussion regarding the uniqueness of the population and how it may have impacted the results (e.g. activity level and dietary habits of farmers) and generalizability of the findings is warranted.
Author Response
Point 1: General—throughout the manuscript, the authors refer to the results in a manner that indicates a causal relationship. Since this is an epidemiological study, please refer to the observed results as associations.
We referred to the observed results as associations throughout the manuscript as suggested.
Point 2: Methods—The statistical analysis section states that age, height, and population group were the only covariates accounted for in the basic model. Items such as education level, smoking status, physical activity, etc. may all be confounders. It is recommended that these, and other relevant variables, be added to the models to determine if an independent association still remains.
We revised the statistical analysis section and have included measures of percent time in moderate plus vigorous activity and average caloric intake in all the baseline and longitudinal analyses since they were associated with at least one of the outcome variables. Smoking status was not associated with any of the outcome variables and was therefore not included in any of the statistical models. The only difference in the findings was with the significance of longitudinal changes in appendicular fat becoming non-significant (p=0.06 rather than p=0.05). We have incorporated these changes into the revised manuscript.
Point 3: Methods/Discussion—The Discussion is the first mention of why the statistical analyses and models did not include body mass index (BMI). However, it is recommended that its exclusion be acknowledged in the Methods section with a brief reason for the exclusion.
We added the following statement to the Methods section: “Due to issues with multicollinearity and the problem of body composition measures being components of both BMI and weight, neither BMI nor weight were included as covariates.”
Point 4: Discussion—the study population is unique and likely not representative of the general U.S. population. A brief discussion regarding the uniqueness of the population and how it may have impacted the results (e.g., activity level and dietary habits of farmers) and generalizability of the findings is warranted.
We included the following statement under limitations: “The majority of the men were farmers who may have different activity patterns and dietary intake than non-farmers, which may influence the relationship between dysglycemia and body composition. However, a study conducted on a representative sample of the U.S. population reported similar findings of an association between dysglycemia and reduced lean mass [10].”
Reviewer 3 Report
In this study, Specker et al studied the association of changes in glycemic status on changes in total body, trunk,and appendicular fat and lean mass in men. I have the following comments:
a. The introduction section - reads well.
b. Methods - adequately explained.
c. Stats - did the sample have sufficient power for this type of analysis... what is the clinically significant changes you have expected to acheive?
d. Covariates - a) info on waist circumference, WHR which is a proxy for body fat needs to be included; b) several diabetes medications results in incresedbody fat accumulation / changes in body fat repository... therefore, please include that in the model; c. also, info on systolic and diastolic blood pressure is missing.
e. discussion - I feel discussion needs to be improved a lot. Its lengthy and not focussed. Please shorten it. Also, please include sample size as one of the limitations.
Author Response
Point 1: Did the sample have sufficient power for this type of analysis... what is the clinically significant changes you have expected to achieve?
We had sufficient power to detect differences in changes in TB and appendicular LM consistent with our hypotheses. However, the hypothesis that would be the most difficult to detect, due to the small sample sizes, would be among men who were pre-diabetic at baseline and whether there was a difference in body composition changes between those who reverted to normoglycemia and those who remained pre-diabetic. We speculated that men who reverted to normoglycemia would have had lower gains in TB and trunk FM and less loss in TB and trunk LM than men who remained pre-diabetic (see Introduction). TB FM changes were consistent with this hypothesis, but in order to detect differences in the change in TB FM between men who were pre-diabetic at baseline and reverted to normoglycemia and those who were pre-diabetic at both visits, based on the observed mean changes and standard deviation, an alpha=0.05 and 80% power, we would have needed about 72 men per category. We have added this under study limitations.
Point 2: Covariates – info on waist circumference, WHR which is a proxy for body fat needs to be included
We did not measure waist circumference and waist to hip ratio in this study. We used DXA measurements of total, trunk, and appendicular fat mass and lean mass. Although WC and WHR are often used as surrogates for truncal adiposity, the use of DXA measurements of lean and fat mass in the total body, trunk and appendicular regions allowed us to determine whether changes in actual measures of lean or fat mass were associated with dysglycemia rather than estimated trunk adiposity as indicated by WC and WHR. DXA measures of trunk fat have been shown to be highly correlated with intraabdominal fat measured with CT or MRI in adults (Treuth MS, et al. Estimating intraabdominal adipose tissue in women by dual-energy x-ray absorptiometry. AJCN 62:527-532, 1995; Clasey JL, et al. The use of anthropometric and dual-energy x-ray absorptiometry (DXA) measures to estimate total abdominal and abdominal visceral fat in men and women. Obesity 7:256-264, 1999.)
Point 3: Several diabetes medications result in increased body fat accumulation/changes in body fat repository... therefore, please include that in the model
At baseline, we had 12 men who self-reported a previous medical diagnosis of T2D and six men who were diagnosed based on fasting or non-fasting glucose at baseline. At the three-year visit, of the 30 men who were classified as having T2D, twelve reported taking the oral anti-diabetic drug, but the specific name of the drug was not provided. Two men reported taking insulin. We created a dummy variable (prescribed drug versus none) and included in the longitudinal models. This variable was not associated with TB, trunk or appendicular FM or LM and therefore was not included.
Point 4: Info on systolic and diastolic blood pressure is missing.
Systolic and diastolic blood pressure measurements were not made.
Point 5: Discussion - I feel discussion needs to be improved a lot. It's lengthy and not focussed.
We revised the Discussion.
Round 2
Reviewer 1 Report
The authors addressed the raised points and improved them.
Reviewer 2 Report
This resubmitted manuscript is a revised version of the original manuscript, which reported on the association between glycemic status and changes in body composition in adult men from the South Dakota Rural Bone Health Study. Results of the study indicated that body composition was not associated with glycemic status at baseline, but longitudinal changes in glycemia status correlated with changes in total body, trunk and appendicular lean mass (LM) and fat mass (FM). This original manuscript was adequately written overall, and the authors have satisfactorily addressed all of the recommended edits and comments.
Reviewer 3 Report
All of my comments were addressed sufficiently.